# Genome-Wide Identification of the Trihelix Transcription Factor Family and Functional Analysis of the Drought Stress-Responsive Genes in *Melilotus albus*

**DOI:** 10.3390/plants12213696

**Published:** 2023-10-26

**Authors:** Qingyan Zhai, Hang Li, Na Wei, Jiyu Zhang, Wenxian Liu

**Affiliations:** State Key Laboratory of Herbage Improvement and Grassland Agro-Ecosystems of Lanzhou University, Key Laboratory of Grassland Livestock Industry Innovation, Ministry of Agriculture and Rural Affairs, Western China Technology Innovation Center for Grassland Industry, Engineering Research Center of Grassland Industry, Ministry of Education, College of Pastoral Agriculture Science and Technology, Lanzhou University, Lanzhou 730000, China; zhaiqy21@lzu.edu.cn (Q.Z.); lih21@lzu.edu.cn (H.L.); wein20@lzu.edu.cn (N.W.)

**Keywords:** drought stress, gene expression analysis, *Melilotus albus*, trihelix transcription factor

## Abstract

The trihelix gene family is a plant-specific family of transcription factors that play an important role in many metabolic pathways, including plant growth and development and stress responses. Drought stress is a major factor limiting the distribution and yield of *Melilotus albus*. However, the distribution of this gene family in *M. albus* and its biological functions in response to drought stress have not been reported. To investigate the responses of functional genes to drought stress in *M. albus*, in this study, a total of 34 *MaGTs* were identified and characterized, of which 32 MaGT proteins were predicted to be nuclear-localized. Based on conserved motif and phylogenetic analyses, the *MaGTs* could be divided into five subgroups (GT-1, GT-2, SH4, GT-γ, SIP1). Seven potential candidate genes for drought tolerance were screened and identified via qRT–PCR based on a transcriptome data analysis of drought stress in *M. albus*. The results indicated that *MaGT17* was not only significantly upregulated in the roots after 24 h of drought stress, but also showed a significant induction in the shoots. This finding further confirms that *MaGT17* is capable of responding to drought stress in *M. albus*. Taken together, these results will offer essential insights for understanding the underlying molecular mechanisms of the trihelix proteins and useful data for further research on the growth, development and stress responses of trihelix proteins in *M. albus*.

## 1. Introduction

Crop yields are inextricably linked to environmental conditions [1]. Drought stress is a major environmental factor that affects the geographical distribution of plants in nature and can lead to reduced crop yields or even crop failure, thereby threatening food security [2]. The negative effects of drought stress will be exacerbated by climate change, which is predicted to increase the frequency of droughts [3]. During the life cycle of a plant, such as during germination or flowering, plants have evolved a number of mechanisms at the morphological, physiological–biochemical and molecular levels to adapt and improve their survival in drought environments, such as reducing the rate of water loss by slowing growth, closing stomata, reducing photosynthesis, accumulating osmoregulatory substances and synthesizing drought-related proteins to improve drought tolerance [4,5,6].

Transcription factors, a class of proteins that bind directly and specifically to upstream sequences of genes, are key molecular switches that protect plants from harsh environments and play a central role in the regulation of plant growth and development and in response to biotic/abiotic stresses [7]. More than 60 families of transcription factors have been identified in plants. The trihelix factor of pea (*Pisum sativum*) was first isolated and identified in the 1990s [8]. The trihelix family is rich in basic amino acids, acidic amino acids, proline and glutamic acids, and its DNA-binding region has three conserved α-helices in the helix–loop–helix–loop–helix conformation; hence, the name trihelix [9,10]. To date, members of the trihelix gene family have been identified in a number of plant genomes, with 30, 41 and 38 trihelix gene members found in Arabidopsis [11], rice (*Oryza sativa*) [12] and *Medicago truncatula* [13], respectively. The trihelix family has been divided into five subfamilies (GT-1, GT-2, GTγ, SH4 and SIP1).

To date, many trihelix proteins have been reported to be involved in various stages of plant growth and development, such as early embryo, stomatal, flower, trichome and seed development [11]. Trihelix genes are also widely involved in regulating various stress responses in plants. *GTL1* loss-of-function analysis in the GT-1 subfamily showed that *A. thaliana gtl1* mutants had a significantly reduced expression. The deletion of the *PTL* gene in the GT-2 subfamily and the resulting *A. thaliana* mutant (*PETAL LOSS*) resulted in pleiotropic phenotypes such as male sterility and the inhibition of sepal growth in *A. thaliana* [14]. The rice pure-mutant strain *osgtγ-1* is more sensitive to salt stress than wild-type rice, and overexpression of *OsGTγ-1* in rice can significantly improve salt tolerance in rice seedlings. Two other members of the GTγ subfamily, OsGTγ-2 and OsGTγ-3, were also induced by most abiotic stresses [15] *A. thaliana ASIL1*, a member of the SIP1 subfamily, targets GT-box-containing embryonic genes and is able to repress the expression of embryonic seed maturation genes in nutritive tissues, thereby inhibiting seed germination and seedling development processes [16]. Several abiotic stress-related genes have been successively reported in the triple-helix family. Overexpression of *GmGT-2A* and *GmGT-2B* found in soybean was able to improve the tolerance of transgenic *A. thaliana* to drought stress [17]. The expression of *BnSIP1-1* protein can be induced by abscisic acid (ABA) and other stresses, such as salt stress and osmotic stress, and overexpression of *BnSIP1-1* significantly increased the germination rate of oilseed rape (*Brassica napus*) seeds under osmotic stress, salt and ABA treatments, suggesting that BnSIP1-1 is involved in ABA synthesis and signaling, and that salt and osmotic stress responses have an important role [18].

*M. albus* is an important forage legume worldwide and is widely used for forage production and soil improvement due to its drought resistance, cold tolerance, infertility tolerance and high-nitrogen-fixation capacity [19,20]. The nutrient content of *M. albus* is relatively high, including crude protein, crude fiber and crude fat, and *M. albus* have a high forage value outdoors [21]. The fodder grass of *M. albus* is rich in nitrogen, phosphorus, potassium, calcium and various micronutrients due to its well-developed root system, rhizomes, roots, stems and leaves and can be used as an herbaceous crop rotation, intercrop or baled green manure to enrich the soil [22]. However, research into the response of *M. albus* to abiotic stress is very limited. With the completion of the whole-genome sequencing of *M. albus*, we have identified and analyzed the *MaGT* gene family at a genome-wide level. Finally, in this study, we identified 34 *M. albus* triple-helix genes using a bioinformatics approach and systematically analyzed their phylogeny, gene structure and expression patterns in response to drought stress. The results of this study may provide a basis for the further functional analysis of drought stress response in *M. albus* and the creation of new germplasm with high drought tolerance through genetic engineering.

## 2. Results

### 2.1. Phylogenetic Categories Analysis of MaGT Genes

To identify the trihelix genes in the *M. albus* genome, we conducted blastp searches on the *M. albus* genome using trihelix sequences documented in *A. thaliana* and *M. truncatula* as query sequences. Using this method, 34 putative *MaGT* genes were identified and submitted to CDD, Pfam and HMM to confirm the GRAS domain. These 34 *Trihelix* genes were divided into five subfamilies with five clades (GT-1, GT-2, SH4, GT-γ and SIP1), of which the SIP1 and SH4 families had the most members with 11 family members each, and the GT-2 family had the fewest members, containing only two. This is consistent with previous studies on *A. thaliana* (30) [11], *M. truncatula* (38) [13] and rice (31) [12]. Among the members of the *M. albus* family, five members contain two GT1 structural domains, and all are located in the SH4 subfamily, and 29 members contain one GT1 structural domain (Figure 1). These results for this subfamily are similar to the gene distribution of different subpopulations of *A. thaliana* [11] and *M. truncatula* [13].

### 2.2. Genomic Organization and In Silico Analysis of the Subcellular Localization of MaGTs

The 34 *MaGT* genes were named according to their physical positions on the chromosomes (from top to bottom). The length of the amino acid sequences encoded by the *MaGT* genes varied from 195 (*MaGT13*) to 1093 (*MaGT22*) amino acids, the relative MW (molecular weight) ranged from 21.82 (*MaGT13*) to 125.16 (*MaGT22*) kDa, the p*I* (isoelectric point) values of the MaGT proteins varied from 4.55 (MaGT20 and MaGT34) to 9.82 (MaGT30) and the GRAVY (grand average of hydropathicity) values ranged from −0.407 (MaGT12) to −1.316 (MaGT20) (Table 1). The predicted subcellular localizations of the MaGT proteins showed that 32 MaGT members might be in the nucleus, while one MaGT protein was anchored in the chloroplast and another was in the cytoplasm (Table 1).

### 2.3. Chromosomal Location Analysis of the MaGT Genes

The *MaGT* genes were mainly concentrated on Chr1, Chr3 and Chr4, with no direct relationship between gene distribution and chromosome length [12]. Chr1 had the highest number of *MaGT* genes (11 genes), and most of them were distributed at the front of the chromosome. Chr2, Chr5 and Chr6 had the fewest *MaGT* genes (two genes) (Figure 2).

### 2.4. Structural Features and Conserved Motifs of the MaGT Genes

The genome sequence and CDS (gene sequences, coding sequences) information of each *MaGT* gene were collected in accordance with the genome information for *M. albus*, and an exon–intron structure diagram was created to help better comprehend the structural evolution of the *MaGT* genes (Figure 3a). Gene structural diversity is an important basis for the evolution of gene families. A single rootless phylogenetic tree was generated from the complete protein sequences of all *MaGT* genes in *M. albus*, and the *MaGT* genes were divided into five subgroups (Figure 3a). The structural results showed that the trihelix family genes generally contained a few introns, including 11 genes without introns (32.4%) and 15 genes with only one intron (44.1%). Lacking in introns represents the main structural forms of the *MaGT* genes. The majority of the SIP1 subfamily members contained fewer than two introns, suggesting that the subfamily is relatively conserved. Otherwise, the SH4, GT-2 and GT-γ subfamilies have no UTR regions (untranslated region), but only introns and exons, which further indicates that these genes are relatively conserved during evolution, although the family is divided into five subfamilies.

To further explore the conserved diversity of the *MaGT* genes, the MEME program predicted the putative motifs and identified different motifs (Figure 3b). We found that most of the members belonging to the same branch have similar motif compositions and exon and intron structures, which indicates that some conserved motifs still exist in the subfamilies during the evolution of the family. As shown in the figure, all MaGT family members contain at least one motif, all proteins contain Motif1 and Motif2, and the majority of family members contain Motif16 (except MaGT1, MaGT7 and MaGT23), which indicates that the family lost at least three motifs (*MaGT11*) during evolution, but there are still subfamilies in the family with some conserved motifs. The C-terminal motifs of the GT-γ subfamily genes and some SH4 subfamily genes contain Motif 8, and the C-terminus of SIP1 branch genes contains Motif 10; a similar phenomenon [23].

### 2.5. Identification of Cis-Acting Elements in the MaGT Gene Promoters

*Cis*-acting elements play very important regulatory roles in the transcriptional initiation of genes, including hormone responses, abiotic stress responses and developmental processes [24]. Using TBtools-II(Toolbox for Biologists)v1.120, the promoters of the *MaGT* genes were extracted from 2000 bp sequences upstream of the start codons in order to explore the function of MaGT promoters. The PlantCARE database was then queried to analyze potential functional elements in the promoter regions of these genes. The promoters are mainly grouped into five categories: (1) seven stress response elements, (2) five hormone synthesis response elements, (3) one cis-acting regulatory element related to meristem expression, (4) one common cis-acting element in promoter and enhancer regions light response element and (5) one light response element (Figure 4). To further investigate the potential functions of the trihelix family, the total of 15 response *cis*-elements in the 2000 bp sequences upstream of the *MaGT* genes were analyzed using the PlantCARE database, including six stress response elements, eight hormone synthesis response elements and one light response element. These *cis*-elements are located upstream of coding sequences and can provide binding sites for transcription factors that respond to abiotic stresses. We found that most of the genes contained *cis*-acting elements in response to stress, such as defense and stress responsiveness, drought inducibility, auxin-responsive element, auxin responsiveness, gibberellin responsiveness and Methyl Jasmonate (MeJA) responsiveness.

### 2.6. Collinearity Gene Analysis of MaGT

To resolve the relationship between gene duplication and genetic evolution of the trihelix genes in *M. albus*, MCScanX and TBtools were used to identify syntenic blocks. An intraspecies covariance analysis was performed to identify six *MaGT* pairs with segmental duplication events (*MaGT30* and *MaGT22*, *MaGT4* and *MaGT33*, *MaGT34* and *MaGT20*, *MaGT3* and *MaGT19*, *MaGT8* and *MaGT17*, *MaGT32* and *MaGT5*) (Figure 5).

We selected three representative species (*A. thaliana* of the Cruciferae family, *M. sativa* of the Leguminosae family and rice of the Gramineae family) for interspecies trihelix family covariance analysis, and the results showed that *MaGT* had different degrees of covariance with the above three species. Among them, it was most genetically close to alfalfa, which is also a legume, with 25 linear gene pairs, and was next most closely related to *A. thaliana*, a crucifer, with 16 linear gene pairs; it was most distantly related to rice, containing only four pairs (Figure 6).

### 2.7. Expression Analysis of the MaGT Genes in M. albus

To clarify the expression patterns of the *MaGT* genes in the shoot (DSS) and root (DSR) during drought stress in *M. albus* and to screen for candidate genes with functions that respond to drought stress, an expression pattern heatmap was constructed based on the RNA-seq data obtained for 34 *MaGT* genes in shoot and root tissues. According to the hierarchical clustering results, these *MaGT* genes had different transcription levels under various drought treatment time points and could be clustered into four clades. The results showed that the roots of most genes were more sensitive to drought stress. The expression of six of these genes (*MaGT3*, *MaGT8*, *MaGT16*, *MaGT17*, *MaGT25* and *MaGT33*) in the roots was upregulated after 3 h of drought stress, and the expression of the first group of branch genes was generally lower or even unexpressed. Among them, the expression of the *MaGT8* and *MaGT17* genes continued to be significantly upregulated in the roots 24 h after being subjected to stress, while the expression of the *MaGT3*, *MaGT7*, *MaGT19* and *MaGT32* genes in the shoots showed a continuous downregulation (Figure 7).

### 2.8. Expression Analysis of MaGT Genes in Response to Drought Stress

To verify the response of the *MaGT* genes to drought stress, seven genes (*MaGT5*, *MaGT6*, *MaGT8*, *MaGT16*, *MaGT17*, *MaGT25* and *MaGT26*) that were significantly upregulated under drought stress were selected and further tested via qRT–PCR (Figure 8). In this study, transcriptome datasets of *M. albus* treated with drought stresses were used to explore the functions of the *MaGT* genes under abiotic stresses. In shoots, all *MaGT* genes were induced under drought stress within 24 h, especially *MaGT17*, whose expression reached peaks at 1 h and whose expression levels were 13.8 times higher than that of the control. In roots, the expression pattern of the *MaGT25* gene in *M. albus* was consistently downregulated within 24 h of drought stress treatment, while all the other genes were upregulated to different degrees, mainly at 3 h and 6 h. Except for the expression of *MaGT25* in the belowground tissues, which was not significantly upregulated, the expression patterns of all other *MaGT* genes were consistent with the RNAseq results. All *MaGT* genes were induced by drought stress to different degrees (Figure 8).

## 3. Discussion

Trihelix genes are widely distributed in plants and are essential for many physiological processes, including plant development. The trihelix family’s evolutionary relationship and genomic identification in various species have been studied thanks to the quick advancement of whole-genome sequencing technologies. To date, studies have been conducted in rice [12], soybean [25], sunflower (*Helianthus annuus* L.) [26] and *M. truncatula* [13]. However, studies on the identification of the trihelix gene family and gene functions in *M. albus* have not been previously reported. Therefore, this present study confirmed and comprehensively analyzed the members of the trihelix family in *M. albus*, which allowed us to study the evolution of the *M. albus* trihelix family and speculate on the biological functions of some unknown *MaGT* genes.

In this study, we identified and characterized a total of 34 *MaGT* genes based on the *M. albus* genome database, which were close in number to those of *A. thaliana* (30) [11], rice (41) [12] and *M. truncatula* (38) [13]. This result also paralleled the genome size of the different species, which shows that there is a positive correlation between the number of trihelix family genes and the genome size of the species. Evolutionary analysis showed that the MaGT proteins fit into five subfamilies. Among the subfamilies, SIP1 and SH4 are the largest subfamilies, with 11 *MaGT* genes each; the GT-2 family is the smallest subfamily, with only two *MaGT* genes. Interestingly, the genes containing two GT1 characteristic structural domains are all distributed in the SIP1 subfamily, while the genes in the other four subfamilies all contain only one GT1 structural domain. Although the genes in this family are relatively nonconserved during the evolutionary process, the conserved structural domains of the genes in this family may still play a central role in the fulfillment of their functions [12]. In addition, the triple-helix DNA-binding structural domain is usually associated with the triple-helix DNA-binding structural domain of MYB/SANT LIKE. The finding that most genes in this family contain the MYB DNA-binding structural domain supports the hypothesis that the triple-helix structural domain may originate from the MYB gene [9,27]. The *MaGT* genes vary in nucleotide sequence length, molecular weight and theoretical isoelectric point. Notably, all *MaGT* genes are very hydrophilic, and most of them are predicted to be localized in the nucleus. This may result in the need for transcription factors to regulate the transcription of downstream genes to adapt to changes in the external environment [28].

Consistent with reports from other species, such as *M. truncatula* (87%) [13], rice (63%) [12] and *Sorghum bicolor* (L.) (86%) [29], the exon–intron structure analysis of the trihelix gene family in *M. albus* shows that most *MaGT* genes lack introns (32.4%) (Figure 3a). The high proportion of intronless genes in the gene family can further indicate the close evolutionary relationship of these family proteins. In addition, the exon–intron pattern analysis of gene families can provide additional insights into their evolution [30]. The conserved motif analysis of *MaGT* genes revealed that the gene family is highly conserved, which indicated that the specific conserved motif may play an important role in the function of a particular cluster.

In general, gene duplication, including tandem, segmental and genomic duplication, affects the generation of novel genes and functional diversity, facilitating the evolution and expansion of gene families in the plant genomes [31]. There were six pairs of *MaGT* genes with segmental duplication in the trihelix gene family of *M. albus*. Compared with tomato (14) [30], *Populus trichocarpa* (14) [32] and soybean (26) [33], the number of *MaGT* genes with duplication events was fewer in *M. albus*. We speculated that most trihelix family genes may have originated from different ancestors and are less conserved in *M. albus*. Additionally, these results indicated that the gene functions of the *M. albus* trihelix family may have a high degree of divergence and diversity. This further indicates that segmental duplication is important to the process of gene duplication in the *MaGT* gene family.

*Cis*-acting regulatory elements are specific motifs located in the promoter regions of genes that act as binding sites for regulatory proteins that can respond to various stresses by regulating the transcription of downstream genes [34]. In addition, some plant hormones, such as abscisic acid, growth hormone, gibberellin and ethylene, are also involved in regulating plant responses to abiotic stresses [35]. Many promoter elements involved in the stress response and hormone response associated with the *MaGT* gene promoter region have been identified, such as TC-rich repeats (*cis*-acting elements involved in defense and stress response), the GARE-motif (gibberellin response element), the TGA-element (growth hormone response element), etc. Various stress response *cis*-acting elements were also identified in the promoters of soybean triple-helix genes [25]. The results indicate that the *MaGT* gene promoter may play an important role in the transcriptional activation of stress response-related genes and the process of the plant stress response.

Studies have shown that triple-helix transcription factors are key regulatory factors in different plant development processes and responses to various stresses, and can be potential candidates for crop improvement [5]. However, the functions of the *MaGT* genes in *M. albus* are still unclear. Therefore, it is necessary to analyze the transcriptional regulation of trihelix transcription factors in *M. albus*, which can be utilized to improve the quality and abiotic stress tolerance of *M. albus*. Here, we first systematically analyzed the expression patterns of these genes under drought stress and identified the potential functions of the *MaGT* genes under drought stress. We observed a significant upregulation of seven *MaGT* genes under drought stress, suggesting that they may be broadly related to the regulation of *M. albus* in response to drought. In particular, *MaGT17* showed upregulation with a 13.8-fold increase. As previously reported, a homolog of *OsGTγ-1*, *MaGT17*, showed a rapid increase in transcript level at 1 h of drought stress treatment. It has been shown that *osgtγ-1* mutants are more sensitive to salt stress and drought stress in rice [15]. *ShCIGT*, a member of the GT1 subfamily, can interact with *SnRK1*, which controls plant metabolism, growth and development, as well as stress tolerance [5]. Based on these data, we conclude that trihelix proteins may also be involved in abiotic stress tolerance.

## 4. Materials and Methods

### 4.1. Plant Materials and Treatments

The variety “Ma46” of *M. albus* was used in this experiment for determining the gene expression profile of the MaGT family in *M. albus* under drought stress. Seeds of *M. albus* were gently broken wall and placed on filter paper to germinate in an artificial climate incubator. After 24 h of germination, the seedlings were transferred into 1/2 MS nutrient solution (pH = 5.8) and grown in an artificial climate incubator (incubation temperature of 25 °C, 14 h of light and 10 h of darkness) for germination. After 30 days, the seedlings were randomly divided into six groups and transferred into 20% PEG-6000 nutrient solution [20]. To reduce the effect of circadian rhythm, all samples were grown in parallel and harvested simultaneously after 24 h. To avoid the influence of circadian rhythm on plant growth and plant gene expression, the shoot and root tissues were harvested after 0 h, 1 h, 3 h, 6 h, 12 h and 24 h of treatment, respectively, and samples were immediately stored in liquid nitrogen at −80 °C. The zero-hour-treated material served as a control.

### 4.2. Identification and Sequence Analysis of the Trihelix Genes in M. albus

We used the MCScanX tool to assess the complete genome sequences and genome annotation files of *M. albus* [19]. Trihelix gene sequences of *A. thaliana* were downloaded from the TAIR website (https://www.arabidopsis.org/tools/bulk/sequences/index.jsp, accessed on 12 December 2021). *M. truncatula* genome information was downloaded from the *M. truncatula* Genome Database (MTGD, http://www.medicagogenome.org/downloads, accessed on 12 December 2021) [13], which was used for searching and identifying trihelix sequences in the *M. albus* genome. All trihelix proteins were identified by the trihelix domain in the Hidden Markov Model (HMM) profile. These model plant protein sequences were used as a query search against *M. albus* in the BLAST 2.9.0 program, and the E value was 10^−10^. The NCBI database was further used to confirm the conserved domains based on the presence of the CDD conserved structural domains (https://www.ncbi.nlm.nih.gov/cdd, accessed on 14 December 2021). The full-length trihelix amino acid sequences derived from 34 *MaGTs* and 30 *AtGTs* were used for phylogenetic analysis. Sequence alignment was performed using Clustal X with the default settings [36]. The neighbor-joining phylogenetic tree was constructed by using MEGA-X software with 1000 bootstrap replicates performed to support statistical reliability.

### 4.3. Chromosomal Distribution and In Silico Gene Prediction

The online tool ProtParam (https://web.expasy.org/protparam/, accessed on 17 December 2021) was used to analyze the molecular weights, grand average of hydropathicity and isoelectric points of *MaGT* genes [37]. The online tool BUSCA (http://busca.biocomp.unibo.it/, accessed on 18 December 2021) was used to predict the subcellular localization of MaGT proteins [38], and GenScript online software (https://www.genscript.com/wolf-psort.html, accessed on 18 December 2021) was used to perform secondary validation of the results [39]. Multiple sequence comparisons of full-length protein sequences were performed using ClustalW2 software [39]. The location information of all *MaGT* genes mounted on eight chromosomes in the *M. albus* genome was extracted and submitted to MCScanX and TBtools-II(Toolbox for Biologists)v1.120. For naming, the prefix ‘Ma’ for *M. albus* was added, followed by a number corresponding to chromosomal position (Figure 2) [40].

### 4.4. Gene Structure, Conserved Motif Identification and Cis-Acting Regulatory Element Analysis

The online software gene structure (http://gsds.gao-lab.org/, accessed on 25 December 2021) was used to analyze the gene structure of the exon–intron structure of *MaGT*. The conserved motif in the full-length MaGT protein was predicted using the online tool MEME (http://meme-suite.org/, accessed on 28 December 2021) [41]. In this study, the *cis*-acting elements in the 2000 bp promoter regions upstream of *MaGT* genes were predicted from *M. albus* data using PlantCARE (http://bioinformatics.psb.ugent.be/webtools/plantcare/html/, accessed on 31 December 2021) [42].

### 4.5. Duplication Analysis

To investigate the chromosomal locations of *MaGT*, gene location data were retrieved from the gff3 annotation of *M. albus*, and covariance blocks between genes within families were calculated using MCScanX software and visualized using TBtools [43]. Based on the downloaded gff files and CDS of *M. truncatula*, *A. thaliana* and rice from the Phytozome database (https://phytozome-next.jgi.doe.gov/, accessed on 5 January 2022), interspecies covariance visualization was performed in TBtools-II(Toolbox for Biologists)v1.120 software to determine distant relatives between species [43].

### 4.6. Transcriptome Analysis for Drought Stress Treatment

To analyze the expression profiles of the *MaGTs* under drought stress in shoots (DSSs) and roots (DRSs), the expression data were obtained as reported previously [20] and used for heatmap creation to display the gene expression profiles with TBtools software.

### 4.7. RNA Isolation and Real-Time Reverse Transcription PCR (qRT–PCR) Analysis

Total RNA was extracted from stress-treated aboveground and belowground tissues of three seedlings from each treatment group at different treatment times using TransZol reagent (DP441) in a mixed plant sampling method. Reverse transcription was performed on total RNA using a cDNA synthesis kit and uniformly diluted to 100 ng·μL^−1^ for qRT–PCR. Primers were designed using Primer3 software to design quantitative primers [44]. The MaActin (*β*-tubulin) gene was used as an internal reference gene. Relative ploidy was calculated using the ∆Ct method, and relative expression levels of each gene were calculated using the 2^−∆∆CT^ method, with the experiment set up as three biological replicates [45].

## 5. Conclusions

In this study, the expression pattern analysis and functional prediction of the *MaGT* gene will help screen potential resistance genes and outline their possible roles in the abiotic stress response of *M. albus*, which will provide a basis for the in-depth analysis of the theory of drought stress response in *M. albus* and the creation of new highly drought-resistant *M. albus* germplasm through genetic engineering. This will lay the foundation for an in-depth analysis for the further functional analysis of the drought stress response in *M. albus* and the creation of new germplasm with high drought resistance through genetic engineering.

## Figures and Tables

**Figure 1 plants-12-03696-f001:**
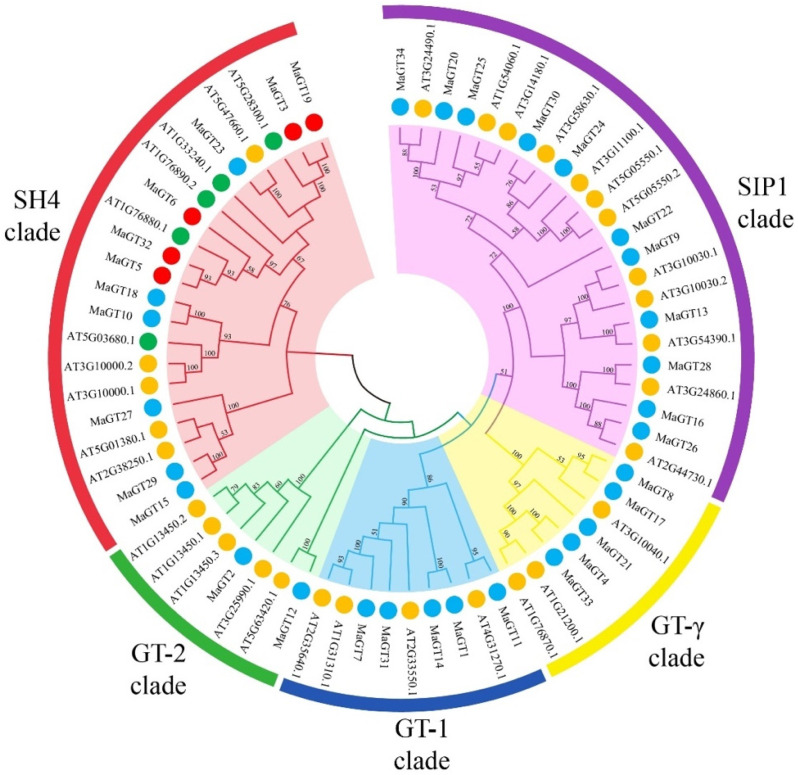
Phylogenetic tree of trihelix proteins in *M. albus* and *A. thaliana*. The branches with different colors represent different subfamilies. The trihelix proteins of *M. albus* are labeled as red and blue circles, respectively; the trihelix proteins of *A. thaliana* are marked as green and yellow circles. The red and green circles indicate trihelix genes with two GT1 domains, and the blue and yellow circles indicate trihelix genes with one GT1 domain.

**Figure 2 plants-12-03696-f002:**
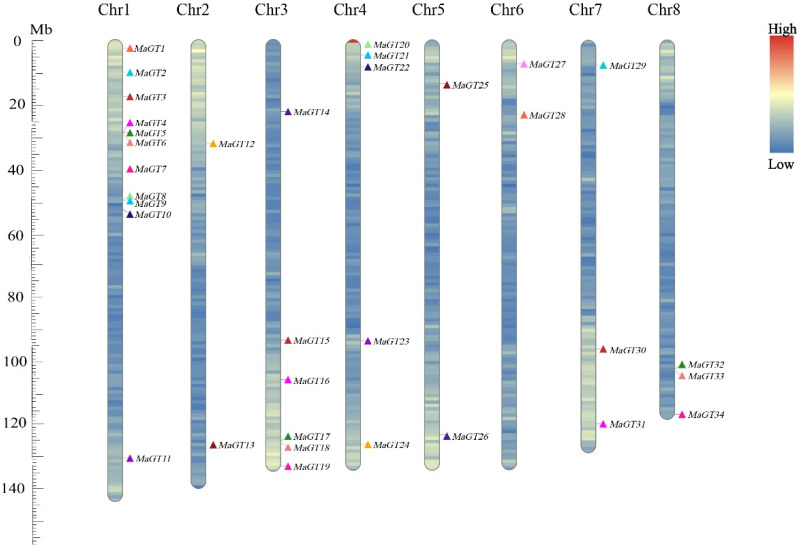
Chromosomal distribution of the nine *MaGT* genes identified in this study. The position of each gene shown on the graph is the average value of the two ends of the gene. The chromosome number is indicated at the top of each bar. The bar on the right side of the image indicates the gene density in the region. The scale on the left is in megabases (Mb).

**Figure 3 plants-12-03696-f003:**
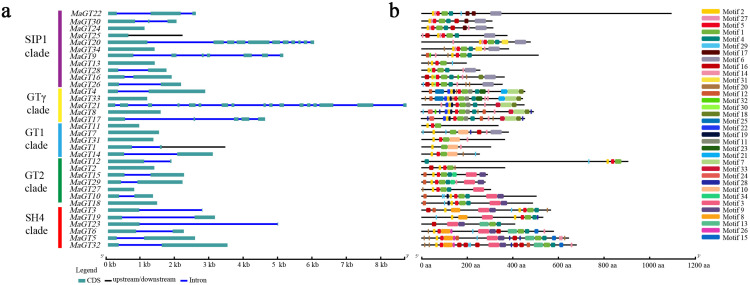
Structure and motif structure of the *MaGT* genes. (**a**) Exon/intron structures of the *MaGT* genes. The exons and introns are represented by green boxes and blue lines, respectively. (**b**) The conserved motifs in the MaGT proteins were identified using MEME. The black lines represent nonconserved sequences. Each motif is indicated by a colored box numbered on the right. The length of the motifs in each protein is shown as a proportion of the length of the amino acids.

**Figure 4 plants-12-03696-f004:**
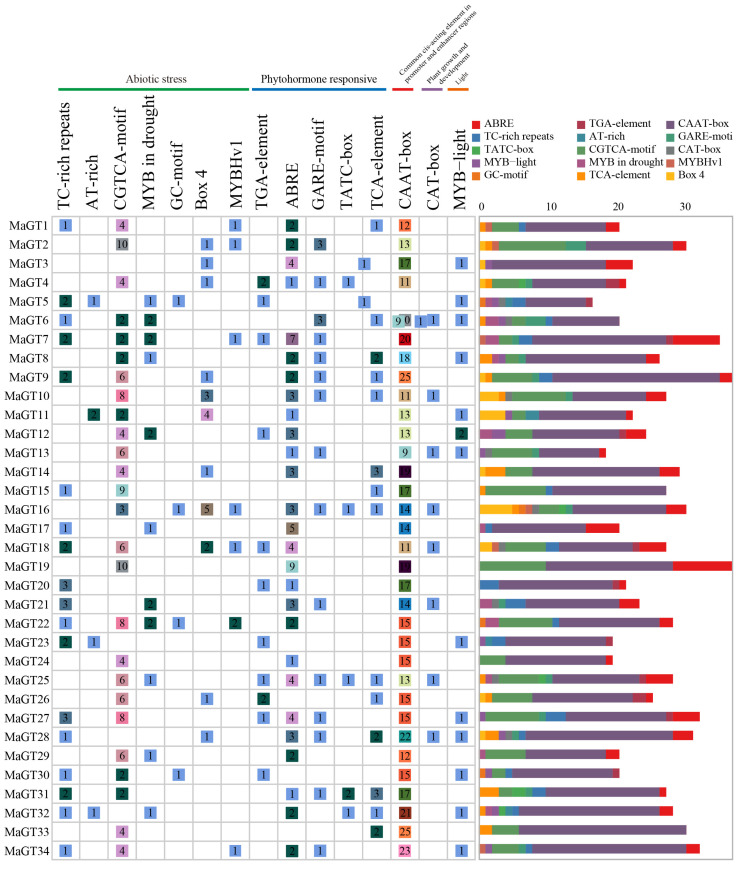
*Cis*-acting elements of the promoter sequences (2000 bp) of 34 *MaGT* genes were analyzed using PlantCARE in *M. albus*. The numbers in the figure’s boxes represent the quantity of cis-acting elements found in the gene. Material synthesis refers to the homeopathic elements involved in the synthesis of substances related to growth and development.

**Figure 5 plants-12-03696-f005:**
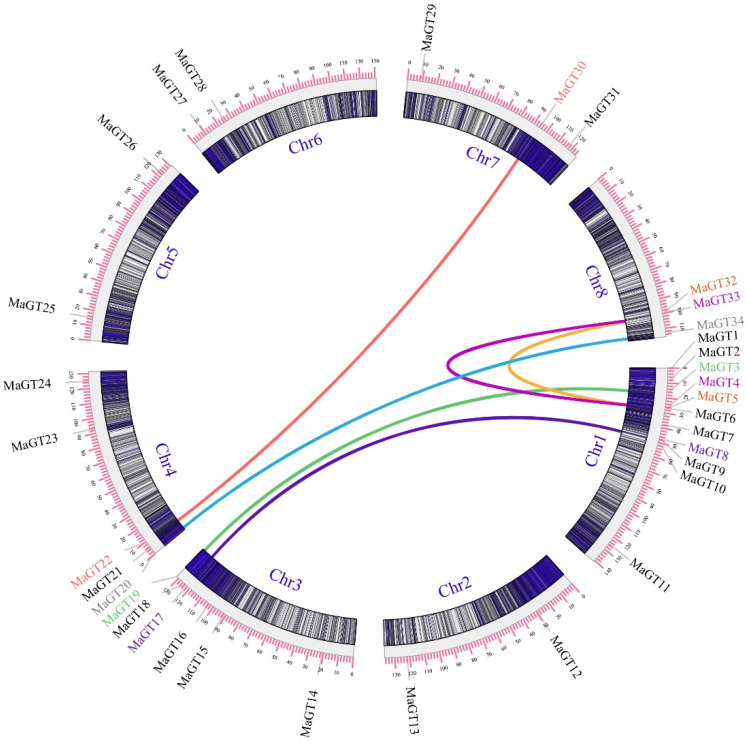
Distribution and synteny analysis of the *MaGT* genes. In the figure, the 8 *M. albus* chromosomes are shown in partial circles, the gene IDs are indicated at the top of each bar, and the collinearity of homologous genes is indicated in the same colors.

**Figure 6 plants-12-03696-f006:**
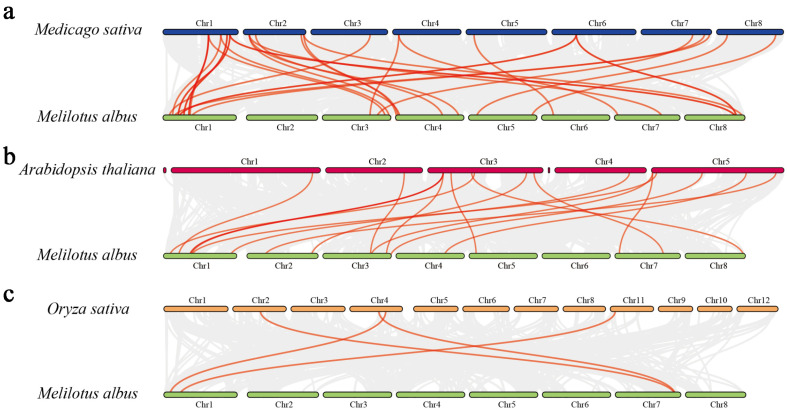
Collinearity analysis of the trihelix genes between *Medicago sativa*, *A. thaliana* and rice: (**a**) *M. sativa*, (**b**) *A. thaliana* and (**c**) rice. The chromosomes of *M. sativa*, *A. thaliana*, rice and *M. albus* are expressed in blue, red, yellow and green, respectively. The red line represents all the collinear gene pairs between the two species. The names of each chromosome are shown above and below.

**Figure 7 plants-12-03696-f007:**
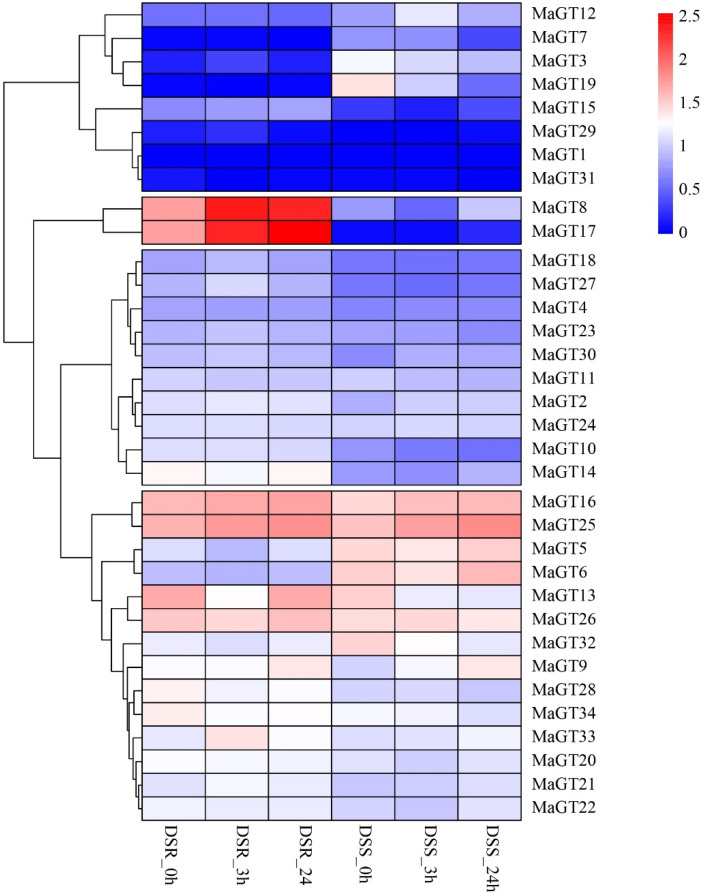
Expression of 34 *MaGT* genes in response to drought treatments. The data were retrieved from transcriptome datasets, and clustering was performed using TBtools. The heatmap shows the relative transcript levels after log2 normalization of the *MaGT* genes under drought stresses. The bar on the right side of the heatmap shows the relative expression values; the more intense the red color, the higher the expression; the more intense the blue color, the lower the expression.

**Figure 8 plants-12-03696-f008:**
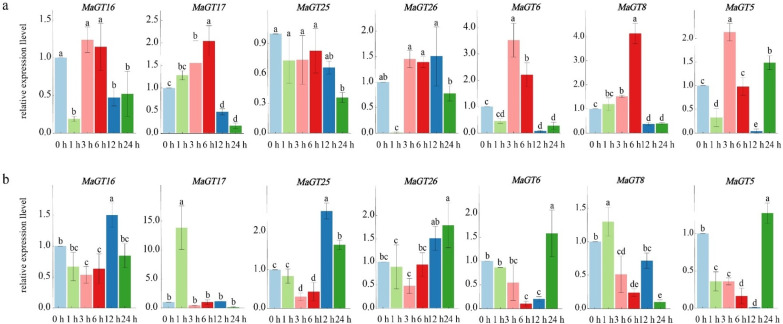
Gene expression analysis of seven *MaGT* genes in *M. albus* roots (**a**) and shoots (**b**) under drought treatment using qRT–PCR; 0 h represents the control. The values shown are the mean ± standard deviation of three replicates. Different letters indicate significant differences between different treatment time (*p* < 0.05).

**Table 1 plants-12-03696-t001:** Details of the MaGT gene family members identified in *M. albus*.

Gene Name	Gene ID	Protein Length (aa)	Protein MW (kDa)	P*I*	Protein GRAVY	Predicted Subcellular Localization
*MaGT1*	*Malbus0100234.1*	301	34.02	4.96	−0.871	nucl
*MaGT2*	*Malbus0100838.1*	363	41.61	5.66	−0.801	nucl
*MaGT3*	*Malbus0101348.1*	564	65.08	5.75	−1.096	nucl
*MaGT4*	*Malbus0102155.1*	450	51.09	6.45	−1.020	nucl
*MaGT5*	*Malbus0102157.1*	643	71.17	5.73	−0.936	nucl
*MaGT6*	*Malbus0102158.1*	577	65.79	6.65	−1.149	nucl
*MaGT7*	*Malbus0102789.1*	379	42.71	9.24	−1.023	nucl
*MaGT8*	*Malbus0103196.1*	491	56.18	6.33	−1.250	nucl
*MaGT9*	*Malbus0103198.1*	510	55.84	6.19	−0.498	nucl
*MaGT10*	*Malbus0103240.1*	551	63.64	6.37	−1.181	nucl
*MaGT11*	*Malbus0105376.1*	334	38.46	5.24	−0.927	nucl
*MaGT12*	*Malbus0202497.1*	903	100.83	8.72	−0.407	nucl
*MaGT13*	*Malbus0205206.1*	195	21.82	9.54	−0.929	nucl
*MaGT14*	*Malbus0300581.1*	252	27.62	6.08	−0.446	nucl
*MaGT15*	*Malbus0302275.1*	287	34.45	6.56	−1.288	nucl
*MaGT16*	*Malbus0302927.1*	361	40.88	9.41	−0.915	nucl
*MaGT17*	*Malbus0304312.1*	452	52.21	6.05	−1.262	nucl
*MaGT18*	*Malbus0304336.1*	485	56.00	6.35	−1.097	nucl
*MaGT19*	*Malbus0305007.1*	530	60.97	6.57	−0.879	nucl
*MaGT20*	*Malbus0400261.1*	475	54.68	4.55	−1.316	nucl
*MaGT21*	*Malbus0400501.1*	447	50.94	6.51	−1.005	nucl
*MaGT22*	*Malbus0400738.1*	1093	125.16	8.87	−0.729	nucl
*MaGT23*	*Malbus0403089.1*	407	45.71	5.92	−0.930	nucl
*MaGT24*	*Malbus0404505.1*	311	35.13	5.12	−0.842	cyto
*MaGT25*	*Malbus0500809.1*	374	40.91	9.66	−0.869	nucl
*MaGT26*	*Malbus0504730.1*	352	38.57	8.90	−0.825	chlo
*MaGT27*	*Malbus0600554.1*	301	35.30	8.67	−1.139	nucl
*MaGT28*	*Malbus0601283.1*	254	29.85	9.20	−0.980	nucl
*MaGT29*	*Malbus0700349.1*	277	33.56	6.44	−1.156	nucl
*MaGT30*	*Malbus0703059.1*	307	34.57	9.82	−0.896	nucl
*MaGT31*	*Malbus0704843.1*	362	40.55	5.57	−0.843	nucl
*MaGT32*	*Malbus0803535.1*	677	75.10	5.81	−1.002	nucl
*MaGT33*	*Malbus0803540.1*	441	50.31	5.98	−1.093	nucl
*MaGT34*	*Malbus0803943.1*	380	43.62	4.55	−1.083	nucl

Note: aa, amino acid; p*I*, isoelectric point; MW, molecular weight; GRAVY, grand average of hydropathicity.

## Data Availability

The data presented in this study are available upon request from the corresponding author.

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
