# Peer review of "Genome-Wide Identification of the Trihelix Transcription Factor Family and Functional Analysis of the Drought Stress-Responsive Genes in Melilotus albus"

_plants, 2023, doi:10.3390/plants12213696_

Round 1

Reviewer 1 Report

Zhai et al conducted a study on the trihelix transcription family in Melilotus albus, covering their phylogeny, structural property, genomic distribution, cis-regulatory elements, intra- and inter- species synteny, as well as responsiveness to drought stress. The study tries to be complete in all aspects, but ends up superficial in most aspects. For example, the cis-acting motif section is rather descriptive, but could be better supported by experiments like Y2H. This renders the conclusions inconclusive or sometimes vague. 

Other points:

Line 117: figure 2: what does color scale (red and blue) mean?

Line 189: figure 4: what does number in each box mean?

Line 201: figure 7: is the color scale (blue and red) absolute expression values or log2 fold changes?

Line 215: text says "MaGT8 and MaGT17 genes continued to be significantly upregulated in the shoots 24h after stress", but I don't see this in Figure 7, please explain.

Author Response

Dear editor and reviewer:

Thank you very much for taking the time to review this manuscript. We appreciate your valuable comments that have help us a lot to improved our manuscript. We have addressed all concerns you pointed out, and hope you will satisfied with this revised version. Our revision details as below.

Thanks again for your time and efforts.

Line 117: figure 2: what does color scale (red and blue) mean?

Response: The bar on the right side of the image represents the gene density. This information has been added and highlighted in lines 125-126.

Line 189: figure 4: what does number in each box mean?

Response: The numbers in the figure's boxes represent the quantity of cis-acting elements found in the gene. This explanation has been added in lines 197-199.

Line 201: figure 7: is the color scale (blue and red) absolute expression values or log2 fold changes?

Response: In figure 7, the color scale (blue and red) represents the absolute log2 fold changes. This information has been added and highlighted in line 211.

Line 215: text says "MaGT8 and MaGT17 genes continued to be significantly upregulated in the shoots 24h after stress", but I don't see this in Figure 7, please explain.

Response: We have re-checked the manuscript carefully and found that MaGT8 and MaGT17 shows a significant rise in the root. We are sorry for this mistake. This mistake has been revised and highlighted in line 225.

Hopefully this revised version will be positively considered for publication.

Wenxian Liu

Lanzhou University

Reviewer 2 Report

This manuscript describes the results of a comprehensive research on the Trihelix transcription factor (TF) gene family members in a N-fixing legume, a valuable honey plant, including analyses of phylogenetic relationships, chromosomal distribution, genome-wide synteny, exon-intron gene structures, conserved motifs and cis-elements, sub-cellular localization, and gene expression patterns in response to PEG-induced drought stress. Bioinformatic tools, transcriptome analysis and qRT-PCR were used. Exploring the TFs at genome-wide scale and their potential biological impact in response to stressors merits attention. In this respect, this systematic study is valuable, the reported results provide a genome-wide glimpse of the leguminous Trihelix gene family and suggest a role of certain members in the plant response to drought stress. My opinion is that the manuscript could be published after minor revision.

The Title corresponds to the manuscript content.

The Abstract is relevant; contains the main results and a conclusion. Please, check the correctness of the sentence which describes the expression of MaGT17 under drought (line 20)

The Introduction is informative.

1)      Lines 80-81:  avoid repetition (genome-wide)

2)      Line 84: consider replacement of the word “theories” (also, in the conclusion part – line 390)

Section Results is well constructed and mostly well-written. The description of the results is supported by one table, and 8 informative figures.  

I have the following remarks/questions/suggestions for this part:

 1)      Figure 1, caption: Please, denote what the circles of different color represent (in both M. albus and A. thaliana)

2)      Figure 2: Please, denote what do the chromosome color from blue to red represent?

3)      Lines 127, 137 and 173: Please explain in brackets the abbreviations CDS, UTR and MeJA.

4)      Line 133: I think that the number of genes without introns is 11 but not 10, based on Fig. 3a (32% of all). Please, check.

5)      Lines 140-143: Please, revise the sentence avoiding the tautology.

6)      Line 144: Please check the phrase “all MaGT family members contain at least one motif”. I was not able to see any gene with one motif on Fig. 3b. The smallest number of motifs is 3 in MaGT11.

7)      Figure 3: mark “a” and “b” on the two sub-figures. In the caption, or below the figure, denote what is CDS. All Figures/Tables should be self-explanatory.

8)      Table 1: Please, define aa, MW, PI, GRAVY. All Figures/Tables should be self-explanatory.

9)      Lines 164-165 and 168-169: Please, revise avoiding the repetitions.

10)  Line 181 and Fig. 6: Medicago sativa or Medicago truncatula? Please, check and correct, if necessary.

11)  Figure 4: What is “Material synthesis”?

12)  Figure 4, caption: why cis-elements were analyzed in 55 MaGT genes?

13)  Figure 7: What does the color difference from blue to red represent?

14)  Text between lines 211 and 217: Please, check the genes that are up-regulated. For example, MaGT8 and MaGT17 are up-regulated in roots not in shoots, based on Fig. 7, if red denotes up-regulation.

15)  Regarding gene MaGT17, there is another confusion: the text on lines 223-225 says “In roots, all MaGT genes were induced under drought stress within 24 h, especially MaGT17, whose expression reached peaks at 1 h and whose expression levels were 13.8 times higher than that of the control”. However, on Fig. 8, the high expression of MaGT17 is presented on “b” sub-figure, i.e. in shoots. Please, check this and correct, if necessary. Also, check the consistency with remark 14 and the Abstract.

Discussion is relatively well-written. Almost all results are discussed, except for the synteny. I have a few minor remarks on this part:

1)      Line 249: the number of identified Trihelix genes in rice is 41 and in M. truncatula is 38 according to the cited references. Please, correct.

2)      Line 249: what is the meaning of “and fewer than those of these plants.”

3)      Lines 269-270: “the exon–intron structure analysis of the trihelix gene family in M. albus shows that most MaGT genes lack introns (79%) (Figure 3A).”

Please, check the number of intronless genes. I was able to count 11 genes without any intron on Fig. 3A, that means 32%, not 79%.

4)      Line 311: osgtγ-1 – A. thaliana or rice? Please, check and correct.

Section Material and Methods is well written.

1)      Line 317: what in Ma46?

The Conclusions part is not informative and contains unnecessary repetitions (in-depth, theory). This part needs entire revision. The most important results should be mentioned.

In summary, I suggest that authors check very carefully the entire manuscript, including the illustrative material – check for correctness and consistency of results description.

English is good. However, at places the meaning of the sentences is unclear. Above, I mentioned a few such cases, but strongly suggest a thorough and careful check of the entire manuscript.

Dear Editors,

This manucsript describes results of a comprehensive study on the trihelix gene family in a leguminous forage species. The used methods are adequate. Potentially interesting outcome is reported relative to the suggested role of some gene members in plant response to drought stress. The manuscript is relatively well-written, needs English improvement, and a careful check of the entire text for correct and consistent description/presentation of the results.

My recommendation is: minor revision.

Thank you for considering me as a reviewer for Plants.

Sincerely,

Svetlana Misheva

Author Response

Dear editor and reviewer:

Thank you very much for taking the time to review this manuscript. We appreciate your valuable comments that have help us a lot to improved our manuscript. We have addressed all concerns you pointed out, and hope you will satisfied with this revised version. Our revision details as below.

Thanks again for your time and efforts.

The Abstract is relevant; contains the main results and a conclusion. Please, check the correctness of the sentence which describes the expression of MaGT17 under drought (line 20)

Response: Thank you for pointing out this mistake in our manuscript. The sentence “The results showed that the expression of trihelix genes was significantly upregulated in the root and shoot of M. albus after 24 h of drought stress, except for MaGT17, whose expression was downregulated in roots, further confirming that trihelix genes are indeed able to respond to drought stress in M. albus.” has been changed as “The results indicated that MaGT17 was not only significantly upregulated in the roots after 24 hours of drought stress but also showed significant induction in the shoots. This finding further confirms that MaGT17 is capable of responding to drought stress in M. albus.” in line 22-25.

Lines 80-81: avoid repetition (genome-wide)

Response: Thanks for your suggestions. The “genome-wide” has been deleted. The sentence “Finally, in this study, we identified 34 M. albus trihelix genes at the genome-wide level using a bioinformatics approach and systematically analyzed their phylogeny, gene structure and expression patterns in response to drought stress.” has been changed as “Finally, in this study, we identified 34 M. albus trihelix genes using a bioinformatics approach and systematically analyzed their phylogeny, gene structure and expression patterns in response to drought stress” in line 84-86.

Line 84: consider replacement of the word “theories” (also, in the conclusion part – line 390)

Response: Thanks for your suggestions. the word “theories” has been replaced with “further functional analysis”. and highlighted in line 87 and line 400.

Section Results is well constructed and mostly well-written. The description of the results is supported by one table, and 8 informative figures. 

I have the following remarks/questions/suggestions for this part:

  • Figure 1, caption: Please, denote what the circles of different color represent (in both albus and A. thaliana)

Response: The red and green circles indicate trihelix genes with two GT1 domains, while the blue and yellow circles indicate trihelix genes with one GT1 domain. The relevant explanatory notes were added and highlighted in line 115-118.

  • Figure 2: Please, denote what do the chromosome color from blue to red represent?

Response: The bar on the right side of the image represents the gene density. This information has been added and highlighted in lines 125-126.

  • Lines 127, 137 and 173: Please explain in brackets the abbreviations CDS, UTR and MeJA.

Response: The “CDS (gene sequences, coding sequences)” has been added on line 134.

The “UTR regions (untranslated region)” has been added in line 145.

The MeJA (Methyl Jasmonate) has been added in line 182.

  • Line 133: I think that the number of genes without introns is 11 but not 10, based on Fig. 3a (32% of all). Please, check.

Response: Thank you for pointing out this mistake in our manuscript. This sentence has been modified and highlighted in line 141.

Lines 140-143: Please, revise the sentence avoiding the tautology.

Response: Thank you for pointing out this mistake in our manuscript. The sentence " implying that these specific motifs are conserved " has been removed in line 149-152.

  • Line 144: Please check the phrase “all MaGT family members contain at least one motif”. I was not able to see any gene with one motif on Fig. 3b. The smallest number of motifs is 3 in MaGT11.

Response: We are sorry for this mistake. The sentence has been modified and highlighted in line 155.

Figure 3: mark “a” and “b” on the two sub-figures. In the caption, or below the figure, denote what is CDS. All Figures/Tables should be self-explanatory.

Response: Thanks for your suggestions. The “a” and “b” have been added to the Figures as required and the full name of CDS has also been supplemented in the manuscript. The sentence has been modified and highlighted in line 134.

Table 1: Please, define aa, MW, molecular weight, PI, GRAVY. All Figures/Tables should be self-explanatory.

Response: Thanks for your suggestions. The define has been added and highlighted in line 120-121.

Lines 164-165 and 168-169: Please, revise avoiding the repetitions.

Response: thanks for your suggestion. The sentence “To further investigate the potential functions of the Trihelix family, the cis-acting elements in the 2,000 bp sequences upstream of the MaGT genes were analyzed using the PlantCARE database, and a total of 15 response elements were predicted” has been changed “To further investigate the potential functions of the Trihelix family, the total of 15 response cis-elements in the 2,000 bp sequences upstream of the MaGT genes were analyzed using the PlantCARE database” in line 174-176

The sentence “These binding sites are located upstream of coding sequences and can provide binding sites for transcription factors that respond to abiotic stresses” has been changed “These cis-elements are located upstream of coding sequences and can provide binding sites for transcription factors that respond to abiotic stresses.” in line 177-179.

  • Line 181 and Fig. 6: Medicago sativa or Medicago truncatula? Please, check and correct, if necessary.

Response: We are sorry for this mistake. We have reconfirmed and modified Medicago truncatula to Medicago sativa. The revised sentence has been highlighted in line 189.

Figure 4: What is “Material synthesis”?

Response: Sorry for the inaccuracy of the term due to an oversight on our part, we have made a further distinction between cis-acting elements in Figure 4. In addition, we have revised this sentence accordingly in line 170-174.

Figure 4, caption: why cis-elements were analyzed in 55 MaGT genes?

Response: Thank you for pointing out this mistake in our manuscript. We have revised the number 55 to 34 as shown in 196.

Figure 7: What does the color difference from blue to red represent?

Response: In figure 7, the color scale (blue and red) represents the absolute log2 fold changes. This information has been added and highlighted in line 211.

Text between lines 211 and 217: Please, check the genes that are up-regulated. For example, MaGT8 and MaGT17 are up-regulated in roots not in shoots, based on Fig. 7, if red denotes up-regulation.

Response: We are so sorry for this mistake. We have reconfirmed and modified “in shoots” to “in roots”. The sentence has been highlighted in line 225.

Regarding gene MaGT17, there is another confusion: the text on lines 223-225 says “In roots, all MaGT genes were induced under drought stress within 24 h, especially MaGT17, whose expression reached peaks at 1 h and whose expression levels were 13.8 times higher than that of the control”. However, on Fig. 8, the high expression of MaGT17is presented on “b” sub-figure, i.e. in shoots. Please, check this and correct, if necessary. Also, check the consistency with remark 14 and the Abstract.

Response: We are so sorry for this mistake. We have reconfirmed and modified “in roots” to “in shoots” in line 233.

Discussion is relatively well-written. Almost all results are discussed, except for the synteny. I have a few minor remarks on this part:

  • Line 249: the number of identified Trihelix genes in rice is 41 and in truncatula is 38 according to the cited references. Please, correct.

Response: Thank you for pointing out this mistake in our manuscript. This part has been revisited and highlighted in line 257-258.

Line 249: what is the meaning of “and fewer than those of these plants.”

Response: We have revisited and corrected the highlighted sentence in line 258-260. The sentence “and fewer than those of these plants” has been removed.

Lines 269-270: “the exon–intron structure analysis of the trihelix gene family in M. albus shows that most MaGT genes lack introns (79%) (Figure 3A).” Please, check the number of intron less genes. I was able to count 11 genes without any intron on Fig. 3A, that means 32%, not 79%.

Response: We are so sorry for this mistake. We have reconfirmed and modified “79%” to “32.4%” in line 279.

  • Line 311: osgtγ-1 – A. thaliana or rice? Please, check and correct.

Response: We are so sorry for this mistake. We have reconfirmed and modified “A. thaliana” to “rice” in line 321.

Section Material and Methods is well written.

  • Line 317: what in Ma46?

Response: Thanks for your suggestions. Ma46 is a species of M. albus. The information “The variety “Ma46” of M. albus was used in this experiment.” has been added and highlighted in lines 328.

Hopefully this revised version will be positively considered for publication.

Wenxian Liu

Lanzhou University

Round 2

Reviewer 1 Report

All my concerns have been adequately addressed.